# GIRAFFEDET: A HEAVY-NECK PARADIGM FOR OBJECT DETECTION

**Yiqi Jiang, Zhiyu Tan, Junyan Wang, Xiuyu Sun\*, Ming Lin, Hao Li**
DAMO Academy, Alibaba Group
{yiqi.jyq, zhiyu.tzy, wangjunyan.wjy}@alibaba-inc.com
{xiuyu.sxy, ming.l, lihao.lh}@alibaba-inc.com

## ABSTRACT

In conventional object detection frameworks, a backbone body inherited from image recognition models extracts deep latent features and then a neck module fuses these latent features to capture information at different scales. As the resolution in object detection is much larger than in image recognition, the computational cost of the backbone often dominates the total inference cost. This heavy-backbone design paradigm is mostly due to the historical legacy when transferring image recognition models to object detection rather than an end-to-end optimized design for object detection. In this work, we show that such paradigm indeed leads to sub-optimal object detection models. To this end, we propose a novel heavy-neck paradigm, GiraffeDet, a giraffe-like network for efficient object detection. The GiraffeDet uses an extremely lightweight backbone and a very deep and large neck module which encourages dense information exchange among different spatial scales as well as different levels of latent semantics simultaneously. This design paradigm allows detectors to process the high-level semantic information and low-level spatial information at the same priority even in the early stage of the network, making it more effective in detection tasks. Numerical evaluations on multiple popular object detection benchmarks show that GiraffeDet consistently outperforms previous SOTA models across a wide spectrum of resource constraints. The source code is available at https://github.com/jyqi/GiraffeDet.

## 1 INTRODUCTION

In the past few years, remarkable progress in deep learning based object detection methods has been witnessed. Despite object detection networks getting more powerful by different designing on architecture, training strategy and so on, the meta-goal that detecting all objects with **large-scale variation** has not been changed. For example, the scale of the smallest and largest 10% of object instances in COCO dataset is 0.024 and 0.472 respectively (Singh & Davis, 2018), which scaling in almost 20 times. This presents an extreme challenge to handle such a large-scale variation by using recent approaches. To this end, we aim to tackle this problem by designing a scale robust approach.

To alleviate the problem arising from large-scale variations, an intuitive way is to use multi-scale pyramid strategy for both training and testing. The work of (Singh & Davis, 2018) trains and tests detectors on the same scales of an image pyramid, and selectively back-propagates the gradients of object instances of different sizes as a function of the image scale. Although this approach improves the detection performance of most existing CNN-based methods, it is not very practical, as the image pyramid methods process every scale image, which could be *computationally expensive*. Moreover, the scale of objects between classification and detection datasets remains another challenge in *domain shift* when using pre-trained classification backbones.

Alternatively, the feature pyramid network is proposed to approximate image pyramids with lower computational costs. Recent methods still rely on superior backbone designing, but *insufficient information exchange* between high-level features and low-level features. For example, some work enhances the entire feature hierarchy with accurate localization signals in lower layers by bottom-up path augmentation, however this bottom-up path design might lack exchange between high-level semantic information and low-level spatial information. According to the above challenges, two questions in this task are raised as follows:

---

\*Corresponding author

- *Is the backbone of the image classification task indispensable in a detection model?*
- *What types of multi-scale representations are effective for detection tasks?*

These two questions motivate us to design a new framework with two sub-tasks two *i.e.,* **efficient feature down-sampling** and **sufficient multi-scale fusion**. First, conventional backbones for scale-sensitive features generation are computationally expensive and exist domain-shift problem. An alternative lightweight backbone can solve these problems. Second, it is crucial for a detector to learn sufficient fused information between high-level semantic and low-level spatial features. According to the above motivations, we design a giraffe-like network, named as **GiraffeDet**, with the following insights: (1) An alternative **lightweight backbone** can extract multi-scale feature transformation without any additional computation costs. (2) A sufficient cross-scale connection, **Queen-Fusion**, like the Queen Piece pathway in chess, to deal with different levels and layers of feature fusion. (3) According to the designed lightweight backbone and flexible FPN, we propose a GiraffeDet family for each FLOPs level. Notably, the experimental results suggest that our GiraffeDet family achieves higher accuracy and better efficiency in each FLOPs level.

In summary, the key **contributions** of our work as follows:
- To the best of our knowledge, we present the first lightweight alternative backbone and flexible FPN combined as a detector. The proposed GiraffeDet family consists of Lightweight S2D-chain and Generalized-FPN, which demonstrates the state-of-the-art performance.
- We design the lightweight space-to-depth chain (S2D-chain) instead of the conventional CNN-based backbone, and controlled experiments demonstrate that FPN is more crucial than conventional backbones in the object detection mode.
- In our proposed Generalized-FPN (GFPN), a novel queen-fusion is proposed as our cross-scale connection style that fuses both level features in previous and current layers, and $\log_2 n$ skip-layer link provides more effective information transmission that can scale into deeper networks.

Based on the light backbone and heavy neck paradigm, the GiraffeDet family models perform well in a wide range of FLOPs-performance trade-offs. In particular, with the multi-scale testing technique, GiraffeDet-D29 achieves 54.1% mAP on the COCO dataset and outperforms other SOTA methods.

## 2 RELATED WORK

It is crucial for the object detector to recognize and localize objects by learning scale-sensitive features. Traditional solutions for the large-scale variation problem mainly based on improved convolutional neural networks. CNN-based object detectors are mainly categorized by two-stage detectors and one-stage detectors. Two-stage detectors (Ren et al., 2015; Dai et al., 2016; He et al., 2017; Cai & Vasconcelos, 2018; Pang et al., 2019) predict region proposals and then refine them by a sub-network, and one-stage detectors (Liu et al., 2016; Lin et al., 2017b; Redmon et al., 2016; Redmon & Farhadi, 2017; Tan et al., 2019; Tian et al., 2019; Zhu et al., 2019; Zhang et al., 2020; 2019; Ge et al., 2021) directly detecting bounding-boxes without the proposal generation step. In this work, we mainly conduct experiments based on one-stage detector methods.

Recently, the main research line is utilizing pyramid strategy, including image pyramid and feature pyramid. The image pyramid strategy is used for detecting instances by scaling images. For example, SNIPER (Singh et al., 2018) propose a fast multi-scale training method, which samples the foreground regions around ground-truth object and background regions for different scale training. Unlike image pyramid methods, feature pyramid methods Lin et al. (2017a); Liu et al. (2018); Chen et al. (2019a); Tan et al. (2020); Sun et al. (2021) fuse pyramidal representations that cross different scales and different semantic information layers. For instance, PANet (Liu et al., 2018) enhances the feature hierarchies on top of feature pyramid network by additional bottom-up path augmentation. Our work focuses on feature pyramid strategy and proposes a sufficient high-level semantic and low-level spatial information fusion method.

Some researchers start working on designing new architectures to solve the large-scale variation problem instead of "backbone-neck-head" architecture in detection tasks. The work of Sun et al. (2019b) proposed the FishNet as an encoder-decoder architecture with skip connections to fuse multi-scale features. SpineNet (Du et al., 2020) designed as a backbone with scale-permuted intermediate features and cross-scale connections that is learned on an object detection task by Neural Architecture Search. Our work is inspired by these methods and proposes a lightweight space-to-depth backbone instead of a CNN-based backbone. However, our GiraffeDet still designed as the

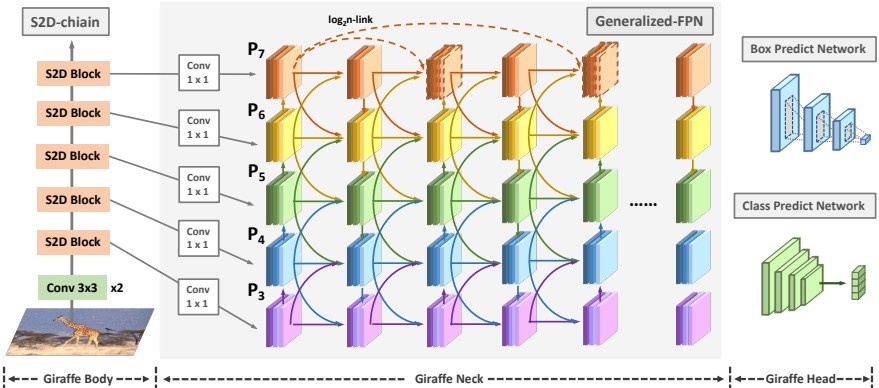

Figure 1: Overview of the GiraffeDet which has three parts: 1) *Body* contains image preprocessing and lightweight S2D-chain;, 2) *Heavy neck* refines and fuses high-level semantic and low-level spatial features; 3) *Head* predicts the bounding box and class label of exist objects.

"backbone-neck-head" architecture. Because this typical architecture is widely used and proved effective in detection tasks.

## 3 THE GIRAFFEDET

Although extensive research has been carried out to investigate efficient object detection, large-scale variation still remains a challenge. To achieve the goal of sufficient multi-scale information exchange efficiently, we proposed the GiraffeDet for efficient object detection, and the "giraffe" consists of lightweight space-to-depth chain, generalized-FPN and prediction networks. The overall framework is shown in Figure 1, which largly follows the one-stage detectors paradigm.

### 3.1 LIGHTWEIGHT SPACE-TO-DEPTH CHAIN

Most feature pyramid networks apply conventional CNN-based networks as the backbone to extract multi-scale feature maps and even learn information exchange. However, recent backbones became much heavier with the development of CNN, it is computationally expensive to utilize them. Moreover, most recent applied backbones are mainly pre-trained on classification dataset, *e.g.,* ResNet50 pre-trained on ImageNet, and we argue these pre-trained backbones are inappropriate in detection task and remains the domain-shift issue. Alternatively, FPN more emphasis on high-level semantic and low-level spatial information exchange. Therefore, we assume that FPN is more crucial than conventional backbones in the object detection model.

Inspired by (Shi et al., 2016; Sajjadi et al., 2018), we propose Space-to-Depth Chain (S2D Chain) as our lightweight backbone, which includes two 3x3 convolution networks and stacked S2D blocks. Concretely, 3x3 convolutions are used for initial down-sampling and introduce more non-linear transformations. Each S2D block consists of a S2D layer and a 1x1 convolution. S2D layer moves spatial dimension information to depth dimension by uniformly sampling and reorganizing features with a fixed gap, so as to down-sample features without additional parameters. Then 1x1 convolutions are used to offer a channel-wise pooling to generate fixed-dimension feature maps. More details are shown in Appendix A.1.

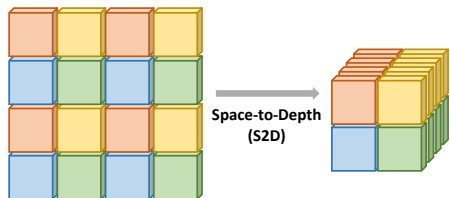

Figure 2: Illustration of the space-to-depth transformation. The S2D operation moves the activation from the spatial dimension to the channel dimension

To verify our assumption, we conduct controlled experiments on different backbone and neck computation ratios in multiple object detection of the same FLOPs in Section 4. The results show that neck is more crucial than conventional backbones in object detection task.

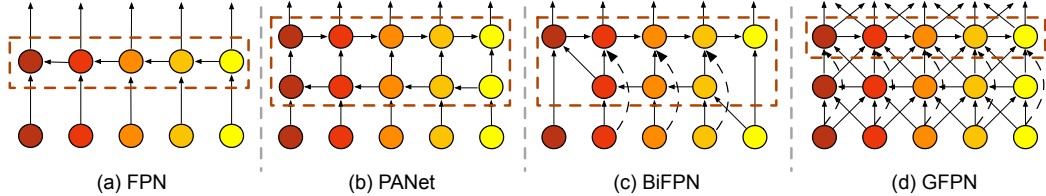

(a) FPN     (b) PANet     (c) BiFPN     (d) GFPN

Figure 3: Feature pyramid network evolution design from level 3 to level 7 (P3 - P7). (a) FPN (Lin et al., 2017a) introduces a top-down pathway to fuse multi-scale features; (b) PANet (Liu et al., 2018) adds an additional bottom-up pathway on top of FPN; (c) BiFPN (Tan et al., 2020) introduces a bidirectional cross-scale pathway; (d) our GFPN contains both queen-fusion style pathway and skip-layer connection. The dashed box refers to the layer in each FPN design.

## 3.2 GENERALIZED-FPN

In feature pyramid network, multi-scale feature fusion aims to aggregate different resolution features that are extracted from the backbone. Figure 3 shows the evolution of feature pyramid network design. Conventional FPN (Lin et al., 2017a) introduces a top-down pathway to fuse multi-scale features from level 3 to 7. Considering the limitation of one-way information flow, PANet(Liu et al., 2018) adds an extra bottom-up path aggregation network, but with more computational cost. Besides, BiFPN (Tan et al., 2020) removes nodes that only have one input edge, and add extra edge from the original input on the same level. However, we observe that previous methods focus only on feature fusion, but lack the inner block connection. Therefore, we design a novel pathway fusion including skip-layer and cross-scale connections, as shown in Figure 3(d).

**Skip-layer Connection**. Compared to other connection methods, skip connections have short distances among feature layers during back-propagation. In order to reduce gradient vanish in such a heavy "giraffe" neck, we propose two feature link methods: dense-link and $log_2n$-link in our proposed GFPN, as shown in Figure 4.

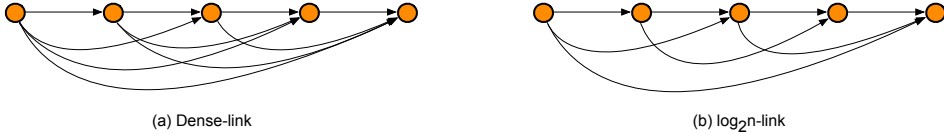

(a) Dense-link              (b) $log_2$n-link

Figure 4: Two link mode of skip-layer connection: (a) dense-link: the concatenation of all preceding layers (b) $log_2$n-link: the concatenation of at most $log_2l + 1$ layers.

• **dense-link**: Inspired by DenseNet (Huang et al., 2017), for each scale feature $P_k^l$ in level $k$, Consequently, the $l^{th}$ layer receives the feature-maps of all preceding layers:

$$P_k^l = Conv(Concat(P_k^0, \ldots, P_k^{l-1})), \tag{1}$$

where $Concat()$ refers to the concatenation of the feature-maps produced in all preceding layers, and $Conv()$ represents a 3x3 convolution.

• **$log_2n$-link**: Specifically, in each level $k$, the $l^{th}$ layer receives the feature-maps from at most $log_2l + 1$ number of preceding layers, and these input layers are exponentially apart from depth i with base 2, as denoted:

$$P_k^l = Conv(Concat(P_k^{l-2^n}, \ldots, P_k^{l-2^1}, P_k^{l-2^0})), \tag{2}$$

where $l - 2^n \geq 0$, $Concat()$ and $Conv()$ also represent concatenation and 3x3 convolution respectively. Compare to dense-link at depth $l$, the time complexity of $log_2n$-link only cost $O(l \cdot log_2l)$, instead of $O(l^2)$. Moreover, $log_2n$-link only increase the short distances among layers during back-propagation from 1 to $1+log_2l$. Hence, $log_2n$-link can scale to deeper networks.

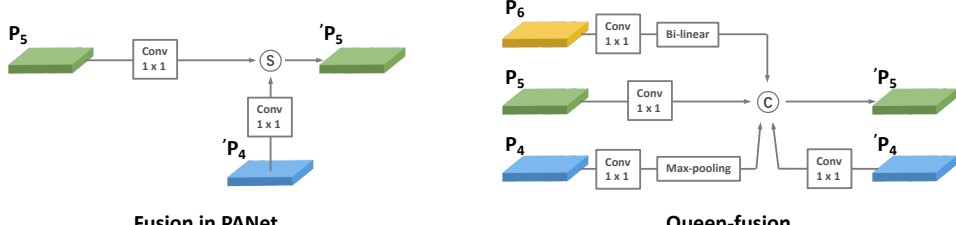

Figure 5: Illustration of cross-scale connection between PANet and our Queen-fusion in GFPN. S and C represent summation and concatenation fusion style, and $'P_k$ denotes node in next layer.

**Cross-scale Connection**. Based on our assumption, our designed sufficient information exchange should contains not only skip-layer connection, but also cross-scale connection, to overcome large-scale variation. Previous works in connecting features between adjacent layers only consider same level feature (Liu et al., 2018) or previous level feature (Tan et al., 2020). Therefore, we propose a new cross-scale fusion named as **Queen-fusion**, that considering both same level and neighbor level features as shown in Figure 3(d), like playing the queen piece in chess. As an example shown in Figure 5(b), the concatenation of Queen-fusion in $P_5$ consists previous layer $P_4$ down-sampling, previous layer $P_6$ up-sampling, previous layer $P_5$ and current layer $P_4$. In this work, we apply *bilinear interpolation* and *max-pooling* as our up-sampling and down-sampling functions respectively.

Therefore, in the extreme large-scale variation scenario, it requires that the model has sufficient high-level and low-level information exchange. Based on the mechanism of our skip-layer and cross-scale connections, the proposed generalized-FPN can be expanded as long as possible, just like the "giraffe neck". With such a "heavy neck" and a lightweight backbone, our GiraffeDet can balance higher accuracy and better efficiency trade-off.

### 3.3 GIRAFFEDET FAMILY

According to our proposed S2D-chain and Generalized-FPN, we can develop a family of different GiraffeDet scaling models that can overcome a wide range of resource constraints. Previous work scale up its detector in the inefficient way, as changing bigger backbone networks like ResNeXt (Xie et al., 2017), or stacking FPN blocks *e.g.,* NAS-FPN (Ghiasi et al., 2019). Specially, EfficientDet (Tan et al., 2020) start using compound coefficient $\phi$ to jointly scale up all dimensions of backbone. Different from EfficientDet, we only focus on the scaling of GFPN layers instead of the whole framework including lightweight backbone. Specifically, we apply two coefficients $\phi_d$ and $\phi_w$ to flexibly scale GFPN depth and width.

Based on our GFPN and eS2D chain, we have developed a GiraffeDet family. Most previous work scale up a baseline detector by changing bigger backbone networks, since their model mainly focus on a single or limited scaling dimensions. As we assume backbone is not critical for object detection task, the GiffeDet family only focus on the scaling of generalized-FPN. Two multipliers are proposed to control the depth (# of layers) and width (# of channels) for GFPN:

$$D_{gfpn} = \phi_d, \ W_{gfpn} = 256 * \phi_w, \quad (3)$$

Following above setting and equation. 3, we have developed six architectures of GiraffeDet, as shown in Table 1. GiraffeDet-D7,D11,D14,D16 have the same level FLOPs with ResNet-series based model and we compare performance of GiraffeDet family with SOTA models in the next section. Note that the layer of GFPN is different with other FPN design as shown in Figure 3. In our proposed GFPN, each layer represents one depth, while the layer of PANet and BiFPN contains two depth.

Table 1: The scaling config for GiraffeDet family — $\phi_d$ is the hyper−parameter that denotes the depth (# of layers) of GFPN. The width (# of channels) of GFPN can be calculated based on $\phi_w$ by Equation 3.

|  | Generalized-FPN | |
|---|---|---|
|  | $\phi_d$ | $\phi_w$ |
| Giraffe-D7 | 7 | 0.7 |
| Giraffe-D11 | 11 | 0.85 |
| Giraffe-D14 | 14 | 0.95 |
| Giraffe-D16 | 16 | 1.0 |
| Giraffe-D25 | 25 | 1.15 |
| Giraffe-D29 | 29 | 1.2 |

# 4 EXPERIMENTS

In this section, we first introduce the implementation details and present our experimental result on the COCO dataset (Lin et al., 2014). Then compare our proposed GiraffeDet family with other state-of-the-art methods, and an in-depth analysis is provided to better understand our framework.

## 4.1 DATASET AND IMPLEMENTATION DETAILS

**COCO dataset**. We evaluate GiraffeDet on COCO 2017 detection dataset with 80 object categories. It includes 115k images for training ($train$), 5k images for validation ($val$) and 20k images with no public ground-truth for testing ($test - dev$). The training of all methods is conducted on the 115k training images. We report results on the validation dataset for ablation study and results of the test-dev dataset from the evaluation server for state-of-the-art comparison and DCN related comparison.

For fair comparison, all results are produced under mmdetection (Chen et al., 2019b) and the standard COCO-style evaluation protocol. GFocalV2 (Li et al., 2021) and ATSS (Zhang et al., 2020) are applied as head and anchor assigner, respectively. Following the the work of (He et al., 2019), all models are trained from scratch to reduce the influence of pre-train backbones on ImageNet. The shorter side of input images is resized to 800 and the maximum size is restricted within 1333. To enhance the stability of scratch training, we adopt multi-scale training for all models, including: 2x imagenet-pretrained (p-2x) learning schedule (24 epochs, decays at 16 and 22 epochs) only in R2-101-DCN backbone experiments, and 3x scratch (s-3x) learning schedule (36 epochs, decays at 28 and 33 epochs) in ablation study, and 6x scratch (s-6x) learning schedule (72 epochs, decays at 65 and 71 epochs) in state-of-the-art comparison. More implementation details in Appendix B.

## 4.2 QUANTITATIVE EVALUATION ON COCO DATASET

We compare GiraffeDet with state-of-the-art approaches in Table. 2. Unless otherwise stated, single-model and single-scale setting with no test-time augmentation is applied. We report accuracy for both test-dev (20k images with no public ground-truth) and val with 5k validation images. We group models together if they have similar FLOPs and compare their accuracy in each group. Notably, model performance depends on both network architecture and training settings. We refer most models from their paper. But for a fair comparison, we also reproduce some of RetinaNet (Lin et al., 2017b), FCOS (Tian et al., 2019), HRNet (Sun et al., 2019a), GFLV2 (Li et al., 2021) with 6x training from scratch, which denoted as †.

**Large-scale Variance**. According to the performance of Figure 6, we can observe that our proposed GiraffeDet achieves the best performance in each pixel scale range, which indicates that the light backbone and heavy-neck paradigm, as well as our proposed GFPN, can effectively solve a large-scale variance problem. Also, under the skip-layer and cross-scale connections, high-level semantic information and low-level spatial information can be sufficiently exchanged. Many object instances are smaller than 1% of the image area in the COCO dataset, making detectors difficult to detect. Even though extremely small instance are difficult to detect, our method still outperforms 5.7% mAP than RetinaNet in the pixel range 0-32, which outperforms the same mAP in the middle pixel range 80-144. Notably, in the scale of pixel range 192-256, the proposed GiraffeDet outperforms the most than other methods, which proves that our design can learn scale-sensitive features effectively.

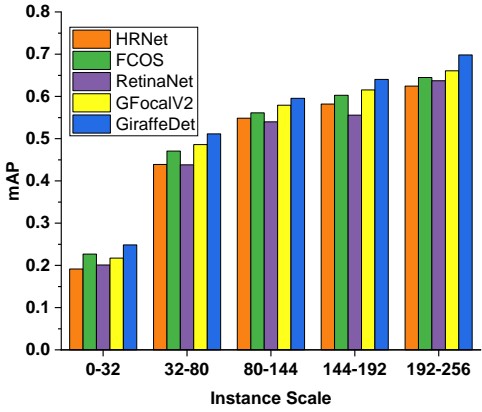

Figure 6: mAP on all scale of object instances (pixels) in five different models under R50 FLOPs level and 6x scratch training, including HRNet (Sun et al., 2019a), GFocalV2 (Li et al., 2021), RetinaNet (Lin et al., 2017b), FCOS (Tian et al., 2019) and our proposed GiraffeDet.

Table 2: **GiraffeDet performance on COCO** - Results for single-model single-scale. test-dev is the COCO test set and val is the validation set. † means that the results are reproduced by 6x scratch training, others are referred from their paper. We group models together if they have similar FLOPs, and compare their accuracy in each group. $\mathbf{MS}_{test}$: multiscale testing, **R:** ResNet, **X:** ResNext, and **W:** low-level features map width in HRNet (# of channels). The head and anchor assigner of GiraffeDet family are GFocalV2 and ATSS.

| Model | #FLOPs Backbone | #FLOPs Neck | #FLOPs total | test-dev $\mathbf{AP_{test}}$ | $\mathbf{AP}_S$ | $\mathbf{AP}_M$ | $\mathbf{AP}_L$ | val-2017 $\mathbf{AP_{val}}$ |
|---|---|---|---|---|---|---|---|---|
| Yolov3-darknet53 | 139.82 | 32.54 | 187.73 | 33.0 | 18.3 | 35.4 | 41.9 | 33.8 |
| RetinaNet-R50† | 76.15 | 16.6 | 229.59 | 40.4 | 23.1 | 43.3 | 52.2 | 40.2 |
| FCOS-R50† | 76.15 | 16.6 | 192.39 | 42.9 | 26.6 | 46.5 | 53.8 | 42.7 |
| GFLV2-R50 | 76.15 | 16.6 | 199.96 | 44.3 | 26.8 | 47.7 | 54.1 | 43.9 |
| GFLV2-R50† | 76.15 | 16.6 | 199.96 | 44.8 | 27.2 | 48.1 | 54.4 | 44.5 |
| HRNetV2p-W18 | 65.13 | 16.34 | 182.06 | 38.3 | 23.1 | 41.5 | 49.2 | 38.1 |
| HRNetV2p-W18† | 65.13 | 16.34 | 182.06 | 40.6 | 25.7 | 43.3 | 50.6 | 40.2 |
| **GiraffeDet-D7** | 3.89 | 76.13 | 186.71 | 45.6 | 28.8 | 48.7 | 55.6 | 44.9 |
| RetinaNet-R101 | 153.74 | 16.6 | 302.62 | 39.1 | 21.8 | 42.7 | 50.2 | 38.9 |
| FCOS-R101 | 153.74 | 16.6 | 265.38 | 41.5 | 24.4 | 44.8 | 51.6 | 40.8 |
| ATSS-R101 | 153.74 | 16.6 | 269.94 | 43.6 | 26.1 | 47.0 | 53.6 | 41.5 |
| PAA-R101 | 153.74 | 16.6 | 269.94 | 44.8 | 26.5 | 48.8 | 56.3 | 43.5 |
| GFLV2-R101 | 153.74 | 16.6 | 272.99 | 46.2 | 27.8 | 49.9 | 57.0 | 45.9 |
| HRNetV2p-W32 | 155.8 | 19.64 | 276.03 | 40.5 | 23.4 | 42.6 | 51.0 | 40.3 |
| HRNetV2p-W32† | 155.8 | 19.64 | 276.03 | 44.6 | 27.9 | 48.0 | 56.8 | 44.1 |
| **GiraffeDet-D11** | 3.89 | 166.73 | 275.39 | 46.9 | 29.9 | 51.1 | 58.4 | 46.6 |
| RetinaNet-R152 | 226.84 | 16.6 | 375.72 | 45.1 | 28.4 | 48.8 | 58.2 | - |
| HRNetV2p-W40 | 229.57 | 21.52 | 351.69 | 42.8 | 27.0 | 46.4 | 54.5 | 42.7 |
| **GiraffeDet-D14** | 3.89 | 251.9 | 361.98 | 47.7 | 30.9 | 51.6 | 60.3 | 47.3 |
| FSAF-X101-64x4d | 304.68 | 16.6 | 421.86 | 42.9 | 26.6 | 46.2 | 52.7 | 42.4 |
| libraRCNN-X101-64x4d | 304.68 | 16.6 | 424.32 | 43.0 | 25.3 | 45.6 | 54.6 | 42.7 |
| FreeAnchor-X101-64x4d | 304.68 | 16.6 | 458.07 | 44.9 | 26.5 | 48.0 | 56.5 | - |
| FCOS-X101-64x4d | 304.68 | 16.6 | 420.87 | 43.2 | 26.5 | 46.2 | 53.3 | 42.6 |
| ATSS-X101-64x4d | 304.68 | 16.6 | 425.43 | 45.6 | 28.5 | 48.9 | 55.6 | - |
| OTA-X101-64x4d | 304.68 | 16.6 | 453.55 | 47.0 | 29.2 | 50.4 | 57.9 | - |
| **GiraffeDet-D16** | 3.89 | 315.69 | 438.59 | 48.7 | 31.7 | 52.4 | 61.3 | 48.3 |
| **GiraffeDet-D25** | 3.89 | 681.02 | 785.4 | 50.5 | 32.2 | 54.2 | 63.5 | 49.9 |
| **GiraffeDet-D29** | 3.89 | 865.44 | 972.97 | 51.3 | 33.1 | 54.9 | 64.9 | 51.0 |
| **GiraffeDet-D29+MS**$_{test}$ | 3.89 | 865.44 | 972.97 | 54.1 | 35.9 | 56.8 | 67.2 | 53.9 |

**Comparison with State-of-the-art Methods**. Table 2 shows that our GiraffeDet family achieves better performance than previous detectors in each same level of FLOPs, which indicates that our method can detect objects effectively and efficiently. 1) Compared to ResNet-based methods on the low-level FLOPs scale, we found that, even if the overall performance is obviously not increased too much, our method has a significant performance in detecting small and large object cases. It indicates our method performs better in large-scale variation dataset. 2) Compared to ResNext-based methods in high-level FLOPs scale, we find that GiraffeDet has a higher performance than in low-level FLOPs slot, which indicates that a good design of FPN can be more crucial than a heavy backbone. 3) Compared to other methods, the proposed GiraffeDet family also has the SOTA performance that proves our design achieves higher accuracy and better efficiency in each FLOPs level. Besides, the NAS-based method consumes a ton of computational resources to cover the search space in the training process, and therefore we do not consider comparing our method with them. Finally, with the multi-scale test protocol, our GiraffeDet achieve 54.1% mAP, especially $AP_S$ increases 2.8% and $AP_L$ increases 2.3% much more than 1.9% in $AP_M$.

## 4.3 ABLATION STUDY

The success of our GiraffeDet can be attributed to both framework design and technical improvements in each component. To analyze the effect of each component in GiraffeDet, we construct ablation studies including: 1) Connection analysis in generalized-FPN; 2) Depth & Width in GFPN; 3) Backbone discussion; 4) GirrafeDet with DCN. More ablation study can be seen in Appendix C.

Table 3: Ablation study on the Connection analysis. The model of "GFPN w/o skip" neck designed without any skip-layer connection, "GFPN-dense" neck model utilizes dense-link and "GFPN-$\log_2$n" neck model utilizes $\log_2$n-link.

| Backbone | Neck | training | FLOPs(G) | $AP_{val}$ | $AP_{50}$ | $AP_{75}$ | $AP_S$ | $AP_M$ | $AP_L$ |
|---|---|---|---|---|---|---|---|---|---|
| S2D chain | stacked FPN | s-3x | 276.11 | 38.8 | 54.3 | 42.2 | 23.1 | 41.9 | 50.6 |
| S2D chain | stacked PANet | s-3x | 275.32 | 40.5 | 56.3 | 43.9 | 24.3 | 43.8 | 52.1 |
| S2D chain | stacked BiFPN | s-3x | 273.1 | 41.0 | 57.1 | 44.3 | 24.0 | 43.6 | 51.9 |
| S2D chain | GFPN w/o skip | s-3x | 273.51 | 41.2 | 57.0 | 44.3 | 25.7 | 43.5 | 51.9 |
| S2D chain | GFPN-dense | s-3x | 273.43 | 41.3 | 57.1 | 44.4 | 26.0 | 43.8 | 52.2 |
| S2D chain | **GFPN-$\log_2$n** | s-3x | 275.39 | 41.8 | 58.1 | 45.7 | 26.4 | 44.9 | 52.7 |

**Connection Analysis**. There are multiple options for constructing pathways between nodes, which are mainly based on graph theory design and human empirical design. Different connections represent different exchanges of information on feature maps. We construct ablation study models and conduct experiments to investigate the effects of our proposed connections. In addition, we stacked basic FPN, PANet and BiFPN several times for fair comparison on the same FLOPs level and used the same backbone and prediction head. Results are given in Table 3.

• *Skip-layer Connection.* According to the results of GFPN-dense and GFPN-$\log_2$n neck of GiraffeDet, we observe that $\log_2$n connection has achieved the best performance, and dense connection only performs slightly better than without any skip-layer connection. It indicates that the $\log_2$n connection provides more effective information transmission from early nodes to later, while dense connection might provides redundant information transmission. Meanwhile, $\log_2$n connection can provides deeper generalized-FPN on the same level of FLOPs. Notably, both generalized-FPN connections obtain higher performance than stacked BiFPN, which can prove that our proposed GiraffeDet can be more efficient.

• *Cross-scale Connection.* From Table 3, we can see that stacked PANet and stacked BiFPN can achieve higher accuracy than their basic structure with bidirectional information flow, which indicates the importance of information exchange in FPN structure. Overall, our GiraffeDet model can achieve better performance, which proves that our Queen-fusion can obtain sufficient high-level and low-level information exchange from previous nodes. Especially, even without skip-layer connection, our generalized-FPN can still outperform other methods.

Table 4: Ablation study on Depth & Width analysis. All models apply S2D-chain as their backbone. "GFPN-$\log_2$n" denotes the GFPN neck utilizes $\log_2$n-link.

| Backbone | Neck | depth | width | training | FLOPs(G) | $AP_{val}$ | $AP_{50}$ | $AP_{75}$ | $AP_S$ | $AP_M$ | $AP_L$ |
|---|---|---|---|---|---|---|---|---|---|---|---|
| S2D chain | stacked FPN | 11 | 307 | s-3x | 274.67 | 38.3 | 55.0 | 41.2 | 22.0 | 42.1 | 51.5 |
| S2D chain | stacked PANet | 11 | 308 | s-3x | 274.41 | 40.6 | 56.9 | 44.3 | 24.2 | 44.1 | 52.0 |
| S2D chain | stacked BiFPN | 11 | 400 | s-3x | 274.51 | 40.5 | 56.8 | 43.9 | 24.1 | 43.6 | 52.0 |
| S2D chain | stacked FPN | 19 | 221 | s-3x | 276.11 | 38.8 | 54.3 | 42.2 | 23.1 | 41.9 | 50.6 |
| S2D chain | stacked PANet | 19 | 221 | s-3x | 275.32 | 40.5 | 56.3 | 43.9 | 24.3 | 43.8 | 52.1 |
| S2D chain | stacked BiFPN | 29 | 221 | s-3x | 273.1 | 41.0 | 57.1 | 44.3 | 24.0 | 43.6 | 51.9 |
| S2D chain | **GFPN-$\log_2$n** | 11 | 221 | s-3x | 275.39 | 41.8 | 58.1 | 45.7 | 26.4 | 44.9 | 52.7 |

**Effect of Depth & Width**. To further fairly comparison with different "Neck", we conduct two groups of experiments comparison with stacked basic FPN, PANet and BiFPN on the same FLOPs level, in order to analysis the effectiveness of depth and width (number of channel) in our proposed generalized-FPN. Note that as shown in Figure 3, each layer of our GFPN and FPN contains one depth, while the layer of PANet and BiFPN contains two depth. As shown in Table 4, we observe that our proposed GFPN outperforms both level of depth and width in all kinds of FPN, which also indicates that the $\log_2$n connection can provide information transmission effectively and the designed Queen-fusion can provide information exchange sufficiently. Moreover, our proposed GFPN can achieve higher performance in a smaller design, as "11" depth and "221" width, which indicates that our design can achieve multi-scale detection efficiently.

**Backbone Effects**. Figure 7 shows the performance of different neck depth and different backbones in the same FLOPs level. The results show that the combination of S2D-chain and GFPN outperforms other backbone models, which can verify our assumption that FPN is more crucial and conventional backbone would not improve performance as depth increasing. In particular, we can observe that performance even decreases with the growth of the backbone model. We consider this might be because the *domain-shift* problem remains higher in a large backbone, and it also proves our assumption.

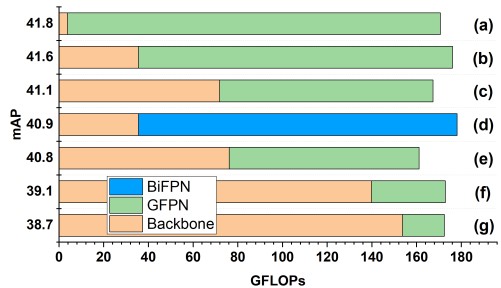

Figure 7: Ablation study on different backbones: a) S2D-chain with GFPN-D11; b) ResNet-18 with GFPN-D10; c) ResNet-34 with GFPN-D8; d) ResNet 18 with stacked BiFPN; e) ResNet-50 with GFPN-D7; f) DarkNet with GFPN-D4; g) ResNet-101 with GFPN-D2.

Table 5: *val*-2017 results of the deformable convolution network applied in GiraffeDet-D11. ‡ denotes the GFPN with synchronized batch normalization (Zhang et al., 2018) for multi-GPU training.

| Backbone | Neck | training | DCN | $AP_{val}$ | $AP_{50}$ | $AP_{75}$ | $AP_S$ | $AP_M$ | $AP_L$ |
|---|---|---|---|---|---|---|---|---|---|
| S2D chain | GFPN-D11 | s-3x | | 41.8 | 58.1 | 45.7 | 26.4 | 44.9 | 52.7 |
| S2D chain | GFPN-D11 ‡ | s-3x | | 42.9 | 59.6 | 46.9 | 27.1 | 46.5 | 54.1 |
| S2D chain | GFPN-D11 ‡ | s-3x | √ | 45.3 | 62.4 | 49.6 | 28.5 | 49.3 | 56.9 |
| S2D chain | GFPN-D11 | s-6x | | 46.6 | 64.0 | 51.1 | 29.6 | 50.8 | 57.9 |
| S2D chain | GFPN-D11 ‡ | s-6x | √ | 49.3 | 66.9 | 53.8 | 31.6 | 53.2 | 61.7 |

Table 6: *val*-2017 results of Res2Net-101-DCN (R2-101-DCN) backbone with multiple GFPN necks. GFPN-*tiny* refers to GFPN of depth as 8 and width as 122 (same FLOPs level as FPN).

| Backbone | Neck | Head | training | $AP_{val}$ | $AP_{50}$ | $AP_{75}$ | $AP_S$ | $AP_M$ | $AP_L$ | FPS |
|---|---|---|---|---|---|---|---|---|---|---|
| R2-101-DCN | FPN | GFLV2 | p-2x | 49.9 | 68.2 | 54.6 | 31.3 | 54.0 | 65.5 | 11.7 |
| R2-101-DCN | GFPN-*tiny* | GFLV2 | p-2x | 50.2 | 68.0 | 54.8 | 32.4 | 54.7 | 65.5 | 11.2 |
| R2-101-DCN | GFPN-D11 | GFLV2 | p-2x | 51.1 | 69.3 | 55.5 | 32.6 | 56.0 | 65.7 | 10.1 |
| R2-101-DCN | GFPN-D11 | GFLV2 | s-6x | 52.3 | 70.2 | 56.7 | 33.9 | 56.8 | 66.9 | 10.1 |

**Results with DCN**

We then conduct experiments to analyse deformable convolution network (DCN)(Dai et al., 2017) in our GiraffeDet, which has been widely used for improving detection performance recently. As shown in Table 5, we observe that DCN can significantly improve the performance of our GiraffeDet. Especially, according to Table 2, GiraffeDet-D11 with DCN can achieve a better performance than GiraffeDet-D16. Also under acceptable inference time, we observe that such a shallow GFPN (*tiny*) with a strong DCN backbone can improve the performance, and the performance has been largely increased with the growth of GFPN depth, as shown in Table 6. Note that as the design of GFPN, Our GiraffeDet is more suitable for scratch training and has significant improvement.

## 5 CONCLUSION

In this paper, we propose a novel heavy-neck paradigm framework, GiraffeDet, a giraffe-like network, to address the problem of large-scale variation. In particular, GiraffeDet uses a lightweight spatial-to-depth chain as a backbone, and the proposed generalized-FPN as a heavy neck. The spatial-to-depth chain is applied to extract multi-scale image features in a lightweight way, and the generalized-FPN is proposed to learn sufficient high-level semantic information and low-level spatial information exchange. Extensive results manifested that the proposed GiraffeDet family achieves higher accuracy and better efficiency, especially detecting small and large object instances.

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

# A ARCHITECTURE DETAILS

## A.1 S2D CHAIN DESIGN

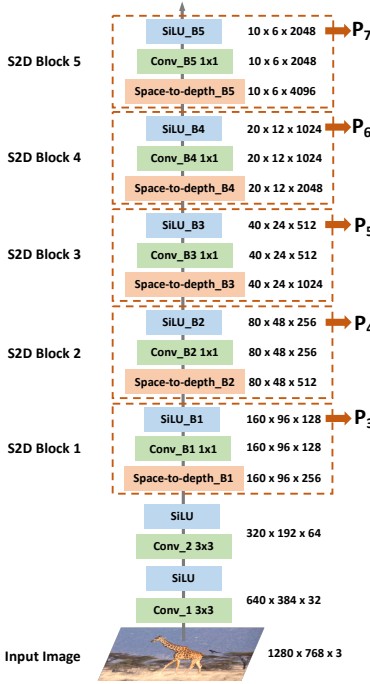

Figure 8: Architecture of Space-to-depth chain. "Conv": convolutional neural networks, "SiLU": sigmoid Linear Units activation function, "Space-to-depth": S2D layer, and "Bx" represents the number of S2D block.

Table 7: Structure of Space-To-Depth chain used in our experiments

| Operation Layer | Number of Filters | Size of Each Filter | Stride Value | Padding Value | Size of Output Feature | FLOPs |
|---|---|---|---|---|---|---|
| Input Image | - | - | - | - | 1280 x 768 x 3 | 0 |
| Convolution Layer | 32 | 3 x 3 x 3 | 2 x 2 | 1 x 1 | 640 x 384 x 32 | 0.21G |
| SiLU Layer | - | - | - | - | 640 x 384 x 32 | 0.21G |
| Convolution Layer | 64 | 3 x 3 x 32 | 2 x 2 | 1 x 1 | 320 x 192 x 64 | 1.34G |
| SiLU Layer | - | - | - | - | 320 x 192 x 64 | 1.34G |
| Space-to-Depth | - | - | - | - | 160 x 96 x 256 | 1.34G |
| Convolution Layer | 128 | 1 x 1 x 256 | 1 x 1 | 0 x 0 | 160 x 96 x 128 | 1.85G |
| SiLU Layer | - | - | - | - | 160 x 96 x 128 | 1.85G |
| Space-to-Depth | - | - | - | - | 80 x 48 x 512 | 1.85G |
| Convolution Layer | 256 | 1 x 1 x 512 | 1 x 1 | 0 x 0 | 80 x 48 x 256 | 2.35G |
| SiLU Layer | - | - | - | - | 80 x 48 x 256 | 2.35G |
| Space-to-Depth | - | - | - | - | 40 x 24 x 1024 | 2.35G |
| Convolution Layer | 512 | 1 x 1 x 1024 | 1 x 1 | 0 x 0 | 40 x 24 x 512 | 2.85G |
| SiLU Layer | - | - | - | - | 40 x 24 x 512 | 2.85G |
| Space-to-Depth | - | - | - | - | 20 x 12 x 2048 | 2.85G |
| Convolution Layer | 1024 | 1 x 1 x 2048 | 1 x 1 | 0 x 0 | 20 x 12 x 1024 | 3.36G |
| SiLU Layer | - | - | - | - | 20 x 12 x 1024 | 3.36G |
| Space-to-Depth | - | - | - | - | 10 x 6 x 4096 | 3.36G |
| Convolution Layer | 2048 | 1 x 1 x 4096 | 1 x 1 | 0 x 0 | 10 x 6 x 2048 | 3.86G |
| SiLU Layer | - | - | - | - | 10 x 6 x 2048 | 3.86G |

## A.2 GENERALIZED FPN DESIGN

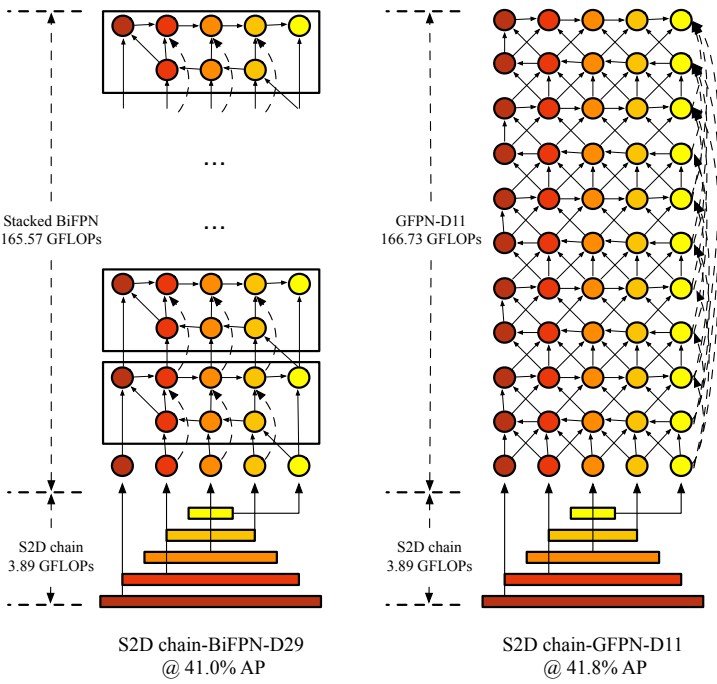

Figure 9: Architecture comparison of stacked BiFPN and our proposed GFPN-D11.

## B MORE IMPLEMENTATION DETAILS

Table 8: List of hyperparameters used.

| Hyperparameter | Value |
|---|---:|
| Batch Size per GPU | 2 |
| Optimizer | SGD |
| Learning Rate | 0.02 |
| Step Decrease Ratio | 0.1 |
| Momentum | 0.9 |
| Weight Decay | $1.0 \times 10^{-4}$ |
| Input Image Size | [1333, 800] |
| Multi-Scale Range (Ablation Study) | [0.8, 1.0] |
| Multi-Scale Range (SOTA) | [0.6, 1.2] |
| GFPN Input Channels | [128, 256, 512, 1024, 2048] |
| GFPN Output Channels | [256, 256, 256, 256, 256] |
| Training Epochs (Ablation Study) | 36 epochs from scratch (decays at 28 and 33 epochs) |
| Training Epochs (SOTA) | 72 epochs from scratch (decays at 65 and 71 epochs) |

## C    MORE ABLATION STUDIES

### C.1    FEATURE FUSION METHODS

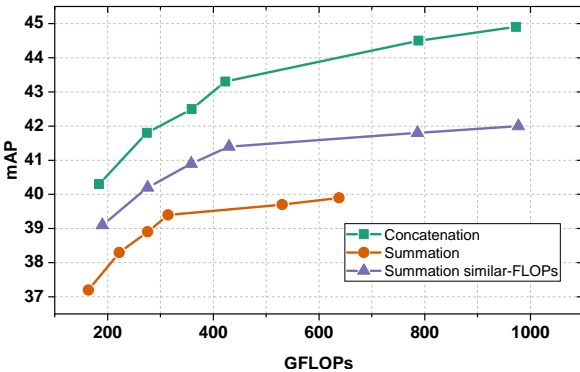

Figure 10: Ablation study on Fusion-style analysis consists three models: 1) "Concatenation" model: GiraffeDet utilizes concatenation fusion style; 2) "Summation" model: GiraffeDet utilizes summation fusion style; 3) "Summation smilar-FLOPs" model: same FLOPs level with "Concatenation" model.

Figure 10 shows the performance of using summation-based feature fusion and concatenation-based feature fusion style. We can observe that the concatenation-based fusion style of features can achieve better performance in the same FLOPs level. Although summation-based feature fusion has fewer FLOPs than concatenation-based style, performance is significantly lower. We think it is not worth sacrificing mAP to have fewer FLOPs. Notably, the performance of the "Summation" model is growing slightly after GFLOPs over 300, which indicates the concatenation-based feature fusion style can be more accurate and efficient again.

### C.2    INFERENCE TIME

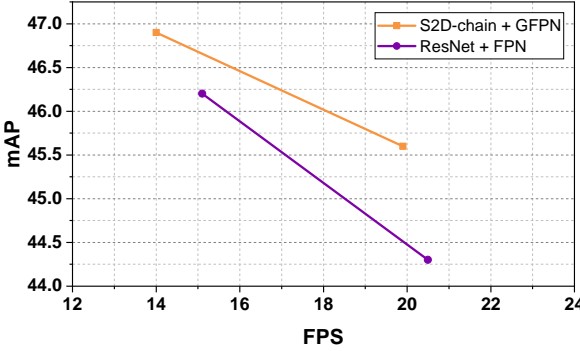

Figure 11: Inference time comparison between "ResNet + FPN" model and "S2D-chain + GFPN" model on the same FLOPs level. Orange line denotes "S2D-chain + GFPN" and purple line denotes "ResNet + FPN".

Table 9: Comparison on inference time between "ResNet + FPN" model and "S2D-chain + GFPN" model on the same FLOPs level.

| Backbone | Neck | Head | training | FLOPs(G) | $AP_{test}$ | $AP_{50}$ | $AP_{75}$ | $AP_S$ | $AP_M$ | $AP_L$ | FPS |
|---|---|---|---|---|---|---|---|---|---|---|---|
| ResNet-50 | FPN | GFLV2 | p-2x | 199.96 | 44.3 | 62.3 | 48.5 | 26.8 | 47.7 | 54.1 | 20.5 |
| S2D chain | GFPN-d7 | GFLV2 | s-6x | 183.67 | 45.6 | 62.7 | 49.8 | 28.8 | 48.7 | 57.6 | 19.9 |
| ResNet-101 | FPN | GFLV2 | p-2x | 272.99 | 46.2 | 64.3 | 50.5 | 27.8 | 49.9 | 57.0 | 15.1 |
| S2D chain | GFPN-d11 | GFLV2 | s-6x | 275.39 | 46.9 | 64.3 | 51.5 | 29.9 | 51.1 | 58.4 | 14.0 |

We conduct inference time experiments to compare our GiraffeDet with the basic detection model (ResNet-FPN-GFocalV2) at the same FLOPs level. From Table 9, we can observe that our GiraffeDet achieves significant improvements with acceptable inference time. We think the reason might be that most popular GPUs are friendly for ResNet-based backbone inferences and memory I/O is sensitive to the concatenate-based fusion on GFPN that will affect the inference speed. Notably, according to Figure 11, the performance of our GiraffeDet decreases slower than the standard model with FPS growth.

## C.3    STANDARD BACKBONE

Table 10: *val*-2017 results of standard backbone with stacked BiFPN and proposed GFPN.

| Backbone | Neck | Head | training | FLOPs(G) | $AP_{val}$ | $AP_{50}$ | $AP_{75}$ | $AP_S$ | $AP_M$ | $AP_L$ |
|---|---|---|---|---|---|---|---|---|---|---|
| resnet-18 | stacked BiFPN | GFLV2 | s-3x | 275.71 | 40.8 | 57.1 | 44.3 | 24.0 | 43.6 | 51.9 |
| resnet-18 | GFPN-d9 | GFLV2 | s-3x | 277.05 | 41.3 | 57.7 | 45.0 | 25.0 | 44.2 | 52.8 |
| resnet-18 | GFPN-d11 | GFLV2 | s-3x | 308.64 | 42.1 | 59.0 | 45.8 | 25.3 | 45.3 | 53.5 |
| resnet-18 | GFPN-d14 | GFLV2 | s-3x | 366.20 | 42.9 | 59.6 | 46.9 | 25.8 | 46.3 | 54.7 |

We also conduct experiments on the ResNet-18 backbone. According to Table 10, our proposed GFPN with a standard backbone can increase with the depth of GFPN growth. Our designed GFPN also outperforms BiFPN under the same FLOPs level.

# D   ADDITIONAL QUALITATIVE RESULTS

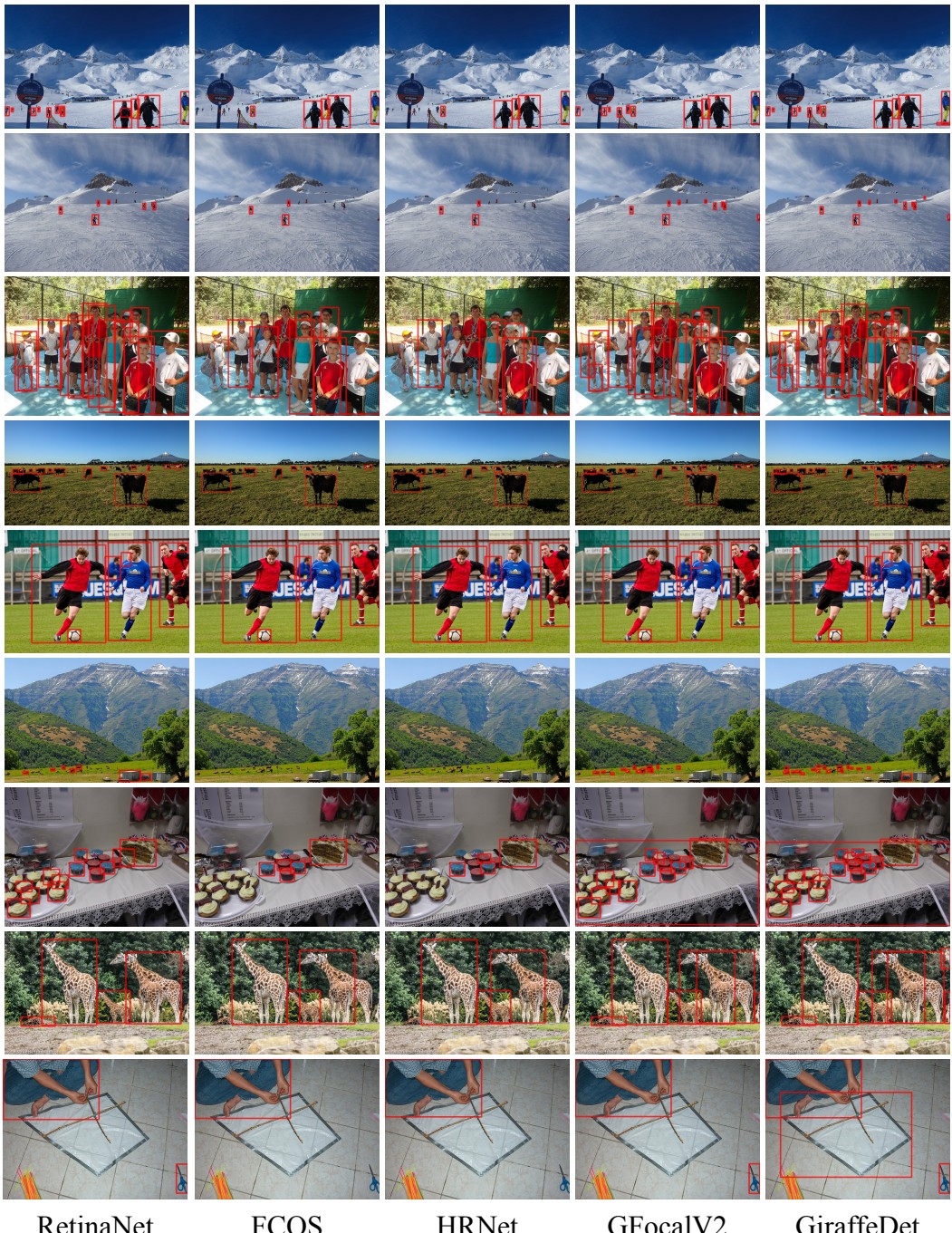

RetinaNet     FCOS     HRNet     GFocalV2     GiraffeDet

Figure 12: Qualitative Evaluation of different approaches for object detection on COCO dataset.

To better illustrate the performance of different approaches, we provide qualitative result in Figure 12. Overall, we can observe that all methods can detect object instances from each image. Furthermore, GiraffeDet can detect more instances than other SOTA methods, especially small object instances, which proves that our designed FPN can be effective in the large-scale variation dataset.

