# OpenReview forum: "GiraffeDet: A Heavy-Neck Paradigm for Object Detection"
_ICLR.cc/2022/Conference — ICLR 2022 Poster_

### Official Review · Reviewer_ygpA · 2021-11-02

**Correctness:** 2
**Technical Novelty And Significance:** 3
**Empirical Novelty And Significance:** 2
**Recommendation:** 8
**Confidence:** 3

**Main Review:**

### Strengths

 - The overall message, an object detector can have a light-weighted bottom-up backbone and a heavy top-down neck is new.
 - The technical contribution of designing the neck in a log-scale connection is neat. It is good to know the log-scale connection performs better than the dense connection.
 - The paper is well written and easy to follow.

### Weaknesses

 - The comparisons in Table. 2 are not clear. What is the detection head of the GiraffeDet? I didn't find the details in the paper. Are they RetinaNet-style anchor-based or FCOS/ ATSS-style single anchor? Ideally, Table 2 should only compare the same detector with different backbones, as the proposed contribution only changes the backbone. It is unclear what to conclude when comparing FCOS with Res50 to a RetinaNet with a different FPN architecture. If the authors are not using the best detection head (or are using a better head than what is listed in Table. 2), then it is necessary to show the proposed backbone can improve the best detection head.

 - At a high level, the paper only shows "using a light backbone and heavy neck" is feasible for object detection, but does not show this is better than the conventional architecture. From Table. 2, the proposed GiraffeDet outperforms the better counterpart in the same block by an ~1mAP. However, given the well-established toolboxes of conventional architecture, it is very hard to convince a user to switch to the proposed architecture. For example, conventional backbones can easily get a performance boost with deformable convolutional layers, while it is unclear how deformable convolutional layers can help the proposed backbones. Ideally, the authors should also compare to DCN-based backbones as they are used in most detection papers. Overall, a 54.1 mAP with test-time augmentation is a victory in Nov 2021.

 - While Table. 2 reports the FLOPs, it is more important to compare the actual inference time on the same machine. I am not sure if the log-scale connection in the proposed framework will make the runtime slower under the same FLOPs.

 - It would be interesting to see how the proposed neck works under a standard backbone. It is fine if the overall FLOPs increase if they significantly outperform BiFPN. Ideally, add the BiFPN entry to Fig 7 (left), with both S2D chain backbone or standard heavy backbone.

 - Fig. 8 is too small to note the difference.



**Summary Of The Paper:**

This paper works on designing network architecture for object detection. While most existing object detectors adopt image classification backbones with a light-weighted "neck" (FPN), this paper proposes the inverse way: using a lightweight backbone and a heavy, dense neck. The proposed network performs ~1 mAP higher than (listed) existing works with similar computations on COCO.

**Summary Of The Review:**

Overall the paper provides an interesting alternative to object detection architectures. However the current experiments do not directly support this claim (mixed change of network architecture and the detection head), and it is unclear if the proposed architecture is compatible with existing techniques (e.g., DCN). The proposed method is also at risk of low run time.

My current rating is a weak reject. In the rebuttal, if the authors show solely replacing the backbone (E.g., replacing ResNeXt-101-DCN-FPN (ideally, the largest EfficientDet or Swin-L) with the proposed network) can improve a state-of-the-art detector (e.g., CascadeRCNN, GFLV2 etc.) with acceptable time cost, I will increase my rating.

---

> ### Author Response · Authors · 2021-11-21
> **Response to Reviewer ygpA (1/2)**
>
> Thanks for all your constructive comments. Please see below our response to the specific questions.
>
> > Q1: The comparisons in Table. 2 are not clear. What is the detection head of the GiraffeDet?
>
> The head and anchor assigner of GiraffeDet family are GFocalV2 and ATSS, respectively. We've made this clear in Section 4.1, and added more training details in Appendix B. Please have a look at the updated version.
>
> > Q2: it is unclear how deformable convolutional layers can help the proposed backbones.
>
> Regarding the deformable convolutional network, we've conducted experiments to analyse deformable convolution network (DCN) applied in our GiraffeDet. As shown in the table below, we observe that DCN can significantly improve the performance of our GiraffeDet. Especially, according to Table 2, GiraffeDet-D11 with DCN can achieve a better performance than GiraffeDet-D16 and close to GiraffeDet-D25.
>
> | model | training | AP_val | AP_50 | AP_75 | AP_s | AP_m | AP_l |
> | --- | --- | --- | --- | --- | --- | --- | --- |
> | GiraffeDet-D11 | s-3x | 41.8 | 58.1 | 45.7 | 26.4 | 44.9 | 52.7 |
> | GiraffeDet-D11 + SyncBN | s-3x | 42.9 | 59.6 | 46.9 | 27.1 | 46.5 | 54.1 |
> | GiraffeDet-D11 + SyncBN + DCN | s-3x | 45.3 | 62.4 | 49.6 | 28.5 | 49.3 | 56.9 |
> | GiraffeDet-D11 | s-6x | 46.6 | 64.0 | 51.1 | 29.6 | 50.8 | 57.9 |
> | GiraffeDet-D11 + SyncBN + DCN | s-6x | 49.3 | 66.9 | 53.8 | 31.6 | 53.2 | 61.7 |
>
> Due to the time reason, we've only explored DCN in our proposed GiraffeDet-D11. We believe that our proposed deeper GiraffeDet can achieve better performance with DCN. We will add the results of deeper GiraffeDet in the camera-ready version.
>
> > Q3: it is more important to compare the actual inference time on the same machine.
>
> Regarding the inference time, we conduct experiments to compare our GiraffeDet with the basic detection model (ResNet-FPN-GFocalV2) at the same FLOPs level. From the table below, we can observe that our GiraffeDet achieves significant improvements with acceptable inference time increasing. As shown in Figure 11 (revised version), the performance of our GiraffeDet decreases slower than the standard model with FPS growth.
>
> | model | FLOPs | AP_test | AP_50 | AP_75 | AP_s | AP_m | AP_l | FPS |
> | --- | --- | --- | --- | --- | --- | --- | --- | --- |
> | r50+FPN+GFLV2 | 199.96 | 44.3 | 62.3 | 48.5 | 26.8 | 47.7 | 54.1 | 20.5 |
> | GiraffeDet-D7 | 183.67 | 45.6 | 62.7 | 49.8 | 28.8 | 48.7 | 57.6 | 19.9 |
> | r101+FPN+GFLV2 | 272.99 | 46.2 | 64.3 | 50.5 | 27.8 | 49.9 | 57.0 | 15.1 |
> | GiraffeDet-D11 | 274.26 | 46.9 | 64.3 | 51.5 | 29.9 | 51.1 | 58.4 | 14.0 |
>
> > Q4: It would be interesting to see how the proposed neck works under a standard backbone. It is fine if the overall FLOPs increase if they significantly outperform BiFPN. Ideally, add the BiFPN entry to Fig 7 (left), with both S2D chain backbone or standard heavy backbone.
>
> Thanks for your kind suggestion. We've revised Figure 7 in the updated version, please have a look. We also conduct experiments on the ResNet-18 backbone. According to the table below, the performance of our proposed GFPN with a standard backbone can increase with the depth of GFPN grows. Our designed GFPN also outperforms BiFPN under the same FLOPs level.
>
> | backbone | neck | head | training | FLOPs | AP_val | AP_50 | AP_75 | AP_s | AP_m | AP_l |
> | --- | --- | --- | --- | --- | --- | --- | --- | --- | --- | --- |
> | resnet-18 | stacked BiFPN | GFLV2 | s-3x | 275.71 | 40.8 | 57.1 | 44.3 | 24.0 | 43.6 | 51.9 |
> | resnet-18 | GFPN-d9 | GFLV2 | s-3x | 277.05 | 41.3 | 57.7 | 45.0 | 25.0 | 44.2 | 52.8 |
> | resnet-18 | GFPN-d11 | GFLV2 | s-3x | 308.64 | 42.1 | 59.0 | 45.8 | 25.3 | 45.3 | 53.5 |
> | resnet-18 | GFPN-d14 | GFLV2 | s-3x | 366.2 | 42.9 | 59.6 | 46.9 | 25.8 | 46.3 | 54.7 |
>
> Due to the time reason, we will add more experiments on other standard backbones in the camera-ready version.

---

> > ### Author Response · Authors · 2021-11-21
> > **Response to Reviewer ygpA (2/2)**
> >
> > > Q5: Fig. 8 is too small to note the difference.
> >
> > Thanks for your kind suggestion. We've revised the qualitative evaluation and moved it to Appendix D, please have a look.
> >
> > > Q6: solely replacing the backbone (E.g., replacing ResNeXt-101-DCN-FPN (ideally, the largest EfficientDet or Swin-L) with the proposed network) can improve a state-of-the-art detector (e.g., CascadeRCNN, GFLV2 etc.) with acceptable time cost.
> >
> > Regarding the strong backbone, we then conduct the experiment of R2-101-DCN with our GFPN, as shown in the table below. The reason we chose Res2Net-101-DCN is that the performance achieves SOTA when applying GFocalV2 head [1]. We observe that such a shallow GFPN (similar Flops with FPN) with a strong DCN backbone can improve performance, and performance has been largely increased with the growth of GFPN depth. Note that the proportion of parameters and FLOPs will count more in the entire detection network with the increase in depth and width of GFPN. Therefore, the R2-101-DCN-based detection network with deep GFPN is more suitable for scratch training and has significant improvements. We have also added this experiment to our updated paper (Section 4.3), please have a look.
> >
> > | backbone | neck | head | training | AP_val | AP_50 | AP_75 | AP_s | AP_m | AP_l | FPS |
> > | --- | --- | --- | --- | --- | --- | --- | --- | --- | --- | --- |
> > | R2-101-DCN | FPN | GFLV2 | pretrained_2x | 49.9 | 68.2 | 54.6 | 31.3 | 54.0 | 65.5 | 11.7 |
> > | R2-101-DCN | GFPN-tiny | GFLV2 | pretrained_2x | 50.2 | 68.0 | 54.8 | 32.4 | 54.7 | 65.5 | 11.2 |
> > | R2-101-DCN | GFPN-D11 | GFLV2 | pretrained_2x | 51.1 | 69.3 | 55.5 | 32.6 | 56.0 | 65.7 | 10.1 |
> > | R2-101-DCN | GFPN-D11 | GFLV2 | scratch_6x | 52.3 | 70.2 | 56.7 | 33.9 | 56.8 | 66.9 | 10.1 |
> >
> > Due to the time reason, we only conducted experiments on R2-101-DCN. We believe that this indicates that our designed GFPN can achieve better performance with a strong backbone and head. We will add more experiments with other strong backbones in the camera-ready version.
> >
> > [1] : Xiang Li, Wenhai Wang, Xiaolin Hu, Jun Li, Jinhui Tang, and Jian Yang. Generalized focal loss v2:Learning reliable localization quality estimation for dense object detection. _In Proceedings of the IEEE/CVF Conference on Computer Vision and Pattern Recognition_, pp. 11632–11641, 2021.

---

> > > ### Comment · Reviewer_ygpA · 2021-12-05
> > > **My concerns are mostly resolved**
> > >
> > > Dear authors,
> > >   Thank you for your informative rebuttal with the additional results. My main concerns in the original reviews, i.e., the fairness of the main comparisons with respect to the detection head and the results under with larger backbone, are resolved in the rebuttal. I am glad to increase my rating to accept based on the new results. Please find my detailed response to each questions below and hope them can help further strengthening the paper:
> > >
> > > Q1: Thank you for the clarification. They are clear now.
> > >
> > > Q2: The results are strong. In the revision, please clarify where the DCN layers are added. Are then only added at the backbone or also at each upsampling layers?
> > >
> > > Q3: Thank you for providing the additional evaluation. The new table makes sense to me.
> > >
> > > Q4: The results make sense. However in the revision, I still encourage the authors to use Res50 as the backbone, as Res50 is the most widely used backbone for detection and there is a common concern of the proposed method about if it makes standard backbones very slow. This is NOT required.
> > >
> > > Q5: Thank you for remaking the figures. A easier way to illustrate might be just provide a zoomed-in region for the small objects (not required).
> > >
> > > Q6: The new results are strong. Thanks!

---

> > > > ### Author Response · Authors · 2021-12-06
> > > > **Thanks for your positive responses**
> > > >
> > > > Dear Reviewer ygpA:
> > > >
> > > > Hope everything is well with you.
> > > >
> > > > Regarding the conducted DCN experiments in Q2, DCN is added at each upsampling layer of GFPN in Table.5, and no DCN in backbone. Meanwhile, DCN-based backbone (R2-101-DCN) is applied in Table.6, and no DCN in FPNs.
> > > >
> > > > Thanks again for your kind suggestion in Q4, Q5, we will update our paper in the published version.
> > > >
> > > > Best regards,
> > > >
> > > > Authors

---

### Official Review · Reviewer_MDN5 · 2021-11-03

**Correctness:** 3
**Technical Novelty And Significance:** 2
**Empirical Novelty And Significance:** 2
**Recommendation:** 5
**Confidence:** 5

**Main Review:**

strengths: The paper had a good exploitation on the connections in skip-layer and cross-scale. The findings are very helpful for the community.
weaknesses: 1. The paper in general does not read well, and more careful proofreading is needed.
                      2. In S2D structure, it is not clear why the number of parameters does not change. If the kernel height/width stay the same, then its depth will increase, resulting in more parameters. I agree the efficiency could be improved since the FLOP is quadratic on activation side length. But in terms of parameters, more details are expected.


**Summary Of The Paper:**

This paper proposes a new network architecture called GiraffeDet. GiraffeDet consists of a light backbone, a heavy neck and a prediction head. In the backbone, the paper proposes the S2D (spatial to depth) structure, claiming its better efficiency. In the neck, the paper proposes denser connections based on FPN, taking cross-scale information exchange into account. The experimental results show the state-of-the-art accuracy compared with other single stage object detection framework on the same FLOP level.

**Summary Of The Review:**

This paper gives another insight into the balance of backbone and neck in the particular problem of object detection. It proposes a framework called giraffeDet, which proves very effective from the experimental results. The main contribution, in my opinion, is about the exploitation on the connections in skip-layer and cross-scale. But this contribution is minor. Also FLOPs is an important metric in efficiency, but the more straightforward inference time will add onto the claim about the light backbone + heavy neck.

---

> ### Author Response · Authors · 2021-11-21
> **Response to Reviewer MDN5**
>
> Thanks for all your constructive comments. Please see below our response to the specific questions.
> > Q1: The paper in general does not read well, and more careful proofreading is needed.
>
> Thanks for your kind suggestion. We've polished our paper and added more details, please have a look at the updated version.
>
> > Q2: In S2D structure, it is not clear why the number of parameters does not change. If the kernel height/width stay the same, then its depth will increase, resulting in more parameters. I agree the efficiency could be improved since the FLOP is quadratic on activation side length. But in terms of parameters, more details are expected.
>
> Regarding the S2D structure,  we've revised Section 3.1 to make the definition clear and added more structure details in Appendix A.1, please have a look. As the scale of the input image is fixed as 1280 x 768 x 3 and the input channels of GFPN is fixed as [128, 256, 512, 1024, 2048], the number of parameters in the S2D chain would not change if the depth of the GFPN was increased. The way in which we use S2D is similar to [1].
>
> [1] Mehdi S. M. Sajjadi, Raviteja Vemulapalli, Matthew Brown; Frame-Recurrent Video Super-Resolution. _Proceedings of the IEEE Conference on Computer Vision and Pattern Recognition (CVPR)_, 2018, pp. 6626-6634
>
> > Q3: Main Contribution is minor.
>
> In this paper, we provide a valuable insight that the backbone is redundant and the neck plays an important role in the object detection task. The controlled experiment result also demonstrates this insight, which is shown in Figure 7. In our approach, we propose a heavy-neck paradigm, which is called GiraffeDet, for the object detection task. Extensive experiments demonstrate that our proposed GiraffeDet outperforms other SOTA methods under the same computation(FLOPs), especially, performance will be significantly increased with the growth of the GFPN depth.
>
> > Q4: Inference time
>
> Regarding the inference time, we conduct experiments to compare our GiraffeDet with the basic detection model (ResNet-FPN-GFocalV2) at the same FLOPs level. From the table below, we can observe that our GiraffeDet achieves significant improvements with acceptable inference time increasing. As shown in Figure 11 (revised version), the performance of our GiraffeDet decreases slower than the standard model, with FPS growth.
>
> | model | FLOPs | AP_test | AP_50 | AP_75 | AP_s | AP_m | AP_l | FPS |
> | --- | --- | --- | --- | --- | --- | --- | --- | --- |
> | r50+FPN+GFLV2 | 199.96 | 44.3 | 62.3 | 48.5 | 26.8 | 47.7 | 54.1 | 20.5 |
> | GiraffeDet-D7 | 183.67 | 45.6 | 62.7 | 49.8 | 28.8 | 48.7 | 57.6 | 19.9 |
> | r101+FPN+GFLV2 | 272.99 | 46.2 | 64.3 | 50.5 | 27.8 | 49.9 | 57.0 | 15.1 |
> | GiraffeDet-D11 | 274.26 | 46.9 | 64.3 | 51.5 | 29.9 | 51.1 | 58.4 | 14.0 |

---

### Official Review · Reviewer_FXm4 · 2021-11-03

**Correctness:** 4
**Technical Novelty And Significance:** 4
**Empirical Novelty And Significance:** 3
**Recommendation:** 8
**Confidence:** 5

**Details Of Ethics Concerns:**

I have no ethics concerns.

**Main Review:**

* Strength
1. The argument in this paper was interesting and the approach (GiraffeDet) to addressing this argument seems to be effective for object detection.
2. Designing the proposed architecture was technically sound.
3. Experiments supports the effectiveness of the proposed method properly.

* Weakness
1. I do not have a big concern on this paper. I recommend to add an experiment comparing the proposed method to baselines when the neck networks are of equal depth.

**Summary Of The Paper:**

This paper raises the question of whether it is effective to conventionally use deep (or heavy) networks trained on ImageNet dataset as backbone of a feature encoder for object detection. Instead, it introduces GiraffeDet, which has a deep neck network that processes multi-scale representations with a shallow backbone. Experiments demonstrate that, given the same memory budget, the proposed method outperforms the conventional method, which rely on the heavy backbone, on MS COCO dataset.

**Summary Of The Review:**

Object detecton accuracy also highly depends on network depth. Although the present experiments well support the effectiveness of the proposed method in terms of efficiency, further experiments comparing methods with various Neck networks sharing depth will provide another perspective demonstrating the advantages of the proposed method.

---

> ### Author Response · Authors · 2021-11-21
> **Response to Reviewer FXm4**
>
> Thanks for your positive comments.
> > Q1: Add an experiment comparing the proposed method to baselines when the neck networks are of equal depth.
>
> Regarding the analysis of depth and width in our GFPN, we conduct two groups of experiments comparison with stacked basic FPN, PANet and BiFPN on the same FLOPs level, as shown in this table. The results of the experiment show that our GFPN outperforms both depth and width levels in all types of FPNs; (Please have a look at more details in Section 4.3.)
>
> | backbone | neck | depth | width | training | FLOPs | AP_val | AP_50 | AP_75 | AP_s | AP_m | AP_l |
> | --- | --- | --- | --- | --- | --- | --- | --- | --- | --- | --- | --- |
> | S2D | stacked FPN | 11 | 307 | s-3x | 274.67 | 38.3 | 55.0 | 41.2 | 22.0 | 42.1 | 51.5 |
> | s2D | stacked PAFPN | 11 | 308 | s-3x | 274.41 | 40.6 | 56.9 | 44.3 | 24.2 | 44.1 | 52.0 |
> | S2D | stacked BiFPN | 11 | 400 | s-3x | 274.51 | 40.5 | 56.8 | 43.9 | 24.1 | 43.6 | 52.0 |
> | S2D | stacked FPN | 19 | 221 | s-3x | 276.11 | 38.8 | 54.3 | 42.2 | 23.1 | 41.9 | 50.6 |
> | s2D | stacked PAFPN | 19 | 221 | s-3x |   275.32 | 40.5 | 56.3 | 43.9 | 24.3 | 43.8 | 52.1 |
> | S2D | stacked BiFPN | 29 | 221 | s-3x |  273.1 | 41.0 | 57.1 | 44.3 | 24.0 | 43.6 | 51.9 |
> | S2D | GFPN | 11 | 221 | s-3x | 274.26 | 41.8 | 58.1 | 45.7 | 26.4 | 44.9 | 52.7 |

---

> > ### Comment · Reviewer_FXm4 · 2021-12-06
> > **I am willing to vote for its acceptance**
> >
> > Thanks for conducting further experiments and adding to the manuscript. I really like the insight given in this paper. I believe that this insight into novel object detection architectures and the ablation studies that support the effectiveness of these architectures could appeal to the ICLR community. Despite reading all the reviewers' concerns, I was very willing to vote for its acceptance.

---

### Author Response · Authors · 2021-11-21
**General Response**

Dear reviewers and meta reviewers,

We appreciate all reviewers for their valuable comments and suggestions. We've revised our manuscript by adding more details and ablation studies as follows:

- We have added a clear description of our designed S2D-chain (Section 3.1 and Appendix A.1);
- We have added a comparison figure between stacked BiFPN and GiraffeDet (Appendix A.2);
- We have revised the description of our designed GirrafeDet family to make the depth and width definition more clear (Section 3.3);
- We have revised Figure 1 and Figure 3 to make the structure more clear;
- We have added more description of our implementation details (Section 4.1 and Appendix B);
- We have added depth and width ablation study (Section 4.3);
- We have added DCN related ablation study (Section 4.3);
- We have revised Figure 7 by adding ResNet-18 + BiFPN result;
- We have added more ablation studies (fusion style, inference time and standard backbone) in Appendix C;
- We have added clear qualitative results in Appendix D.

We will release our code and models in the camera-ready version, and please see below our response to each reviewer.
If you have any questions or suggestions, please put your comments on OpenReview.

---

### Decision · Program_Chairs · 2022-01-20

**Decision:**

Accept (Poster)

**Comment:**

Three experts reviewed the paper. Two reviewers recommended acceptance as they liked that the work identified a legacy design in object detection networks and resolved it by a new module. All reviewers found the empirical results strong. Reviewer MDN5 recommended rejection main for the concern that this newly designed module is a standard exploitation of network architectures. AC sided with the positive reviewers because of the paper's identification of a legacy design in object detection and the strong experimental results. Hence, the decision is to recommend the paper for acceptance. The reviewers did raise some valuable concerns that should be addressed in the final camera-ready version of the paper. The authors are encouraged to make the necessary changes to the best of their ability. We congratulate the authors on the acceptance of their paper!